# Recent Advances in the Field of Artificial Intelligence for Precision Medicine in Patients with a Diagnosis of Metastatic Cutaneous Melanoma

**DOI:** 10.3390/diagnostics13223483

**Published:** 2023-11-20

**Authors:** Hayley Higgins, Abanoub Nakhla, Andrew Lotfalla, David Khalil, Parth Doshi, Vandan Thakkar, Dorsa Shirini, Maria Bebawy, Samy Ammari, Egesta Lopci, Lawrence H. Schwartz, Michael Postow, Laurent Dercle

**Affiliations:** 1Department of Clinical Medicine, Touro College of Osteopathic Medicine, Middletown, NY 10940, USA; lotfalla.andrew@gmail.com (A.L.); mdbebawy@gmail.com (M.B.); 2Department of Clinical Medicine, American University of the Caribbean School of Medicine, 33027 Cupecoy, Sint Maarten, The Netherlands; abanoub.a.nakhla@gmail.com; 3Department of Clinical Medicine, Campbell University School of Osteopathic Medicine, Lillington, NC 27546, USA; david.a.khalil@gmail.com (D.K.); ppdoshi1201@email.campbell.edu (P.D.); v_thakkar1310@email.campbell.edu (V.T.); 4Department of Radiology, Shahid Beheshti University of Medical Sciences, Tehran 1981619573, Iran; dorsa74@gmail.com; 5Département d’Imagerie Médicale Biomaps, UMR1281 INSERM, CEA, CNRS, Gustave Roussy, Université Paris-Saclay, 94800 Villejuif, France; samy.ammari@gustaveroussy.fr; 6ELSAN Département de Radiologie, Institut de Cancérologie Paris Nord, 95200 Sarcelles, France; 7Nuclear Medicine Unit, IRCCS Humanitas Research Hospital, 20089 Rozzano, Italy; egesta.lopci@gmail.com; 8Department of Radiology, New York-Presbyterian, Columbia University Irving Medical Center, New York, NY 10032, USA; lschwartz@cumc.columbia.edu; 9Melanoma Service, Department of Medicine, Memorial Sloan Kettering Cancer Center, New York, NY 10065, USA; 10Weill Cornell Medical College, New York, NY 10065, USA

**Keywords:** metastatic melanoma, artificial intelligence, immunotherapy, radiology

## Abstract

Standard-of-care medical imaging techniques such as CT, MRI, and PET play a critical role in managing patients diagnosed with metastatic cutaneous melanoma. Advancements in artificial intelligence (AI) techniques, such as radiomics, machine learning, and deep learning, could revolutionize the use of medical imaging by enhancing individualized image-guided precision medicine approaches. In the present article, we will decipher how AI/radiomics could mine information from medical images, such as tumor volume, heterogeneity, and shape, to provide insights into cancer biology that can be leveraged by clinicians to improve patient care both in the clinic and in clinical trials. More specifically, we will detail the potential role of AI in enhancing detection/diagnosis, staging, treatment planning, treatment delivery, response assessment, treatment toxicity assessment, and monitoring of patients diagnosed with metastatic cutaneous melanoma. Finally, we will explore how these proof-of-concept results can be translated from bench to bedside by describing how the implementation of AI techniques can be standardized for routine adoption in clinical settings worldwide to predict outcomes with great accuracy, reproducibility, and generalizability in patients diagnosed with metastatic cutaneous melanoma.

## 1. Introduction

Medical imaging plays an essential role in the management of all cancers, especially those diagnosed with malignant cutaneous melanoma, a current public health concern due to the increasing incidence, prevalence, and morbidity/mortality. Immunotherapy and inhibitors that target BRAF (a serine/threonine protein kinase) and MEK (a mitogen-activated protein kinase in the MAPK pathway) are the leading therapeutic strategies used in advanced melanoma. Prior to these strategies, treatments were found to have limited success. While BRAF and MEK inhibitors are only indicated for use in about 40–50% of patients with BRAF V600 mutations, immunotherapies have been shown to be effective regardless of a patient’s BRAF mutation status. Despite these advancements, melanoma is the 17th most common cause of cancer worldwide, and the incidence has rapidly increased over the last 50 years, leading to a large majority of skin cancer deaths [1].

In recent years, an upsurge in artificial intelligence (AI) research in patients with cutaneous melanoma suggests that the AI-based analysis of medical images may enhance a wide range of clinical tasks. It has changed the way providers manage patients with cancer, from the selection of exam/imaging modality to detection, staging, prognostication, treatment planning and delivery, response assessment, and prediction of toxicity.

This review explores the existing literature on CT, MRI, and PET scans and provides an overview of the potential benefits as well as the restrictions of adopting AI, including but not limited to detection/diagnosis, staging, treatment planning and delivery, response assessment, treatment toxicity assessment, and surveillance of malignant cutaneous melanoma. 

In this review, we discuss the potentially promising role of AI for the precision management of patients within the various clinical specialties involved in patient care by exploring various perspectives on using AI-based tools in radiology, immunotherapy, and radiation therapy, thus increasing diversity in future clinical trials.

Finally, we will explain the possible future directions for AI as it relates to cutaneous melanoma. The implementation of AI methods is changing the landscape of medical imaging, both in clinical practice and research [2]. The goal of AI is to develop a multiomics approach by extracting biomarkers and signatures with other parameters via image analysis, potentially resulting in revolutionary personalized care by selecting the most effective treatments with the fewest side effects. AI can study radiographic images over time and evaluate tumor mechanisms of molecular resistance, leading to new therapeutic treatments against melanomas. Hopefully, by implementing standardization, the use of AI could be adopted worldwide and pave the way for reproducible and generalizable tools that will improve care for patients with malignant cutaneous melanoma.

## 2. Concepts in AI

AI is a vast field that encompasses algorithms designed to accomplish tasks traditionally achieved via human intelligence. It includes machine learning (ML) and deep learning (DL), a subset of ML. AI transforms images into numerical data, thus providing information beyond visual interpretation. 

DL algorithms employ complex models with structures similar to neural networks. These algorithms reduce the need for human input and identify patterns automatically based on data [3]. They can learn to predict outcomes without relying on any prior domain-specific knowledge [4]. With more training data, these DL algorithms can form new associations between images and diseases; there is no limit to their capacity to learn [3].

Radiomics is a method that extracts quantitative characteristics from images using data algorithms with features generated using ML algorithms. These algorithms and models can replicate certain input–output relationships in a dataset without prior decision-making principles. Domain experts are required to select these features, which include characteristics such as intensity heterogeneity, edge sharpness, and shape irregularity that are known to be associated with disease [3].

## 3. Challenges of AI

One of the biggest challenges in using automated AI tools for medical decision making is that the models may need to work better across different institutions with diverse patient populations and imaging procedures. Overfitting is a common problem in developing these AI models, which has been partially addressed by improving the training methods. Another significant barrier to implementing automated AI-based decision-making tools is underspecification, which requires a thorough understanding and rectification of concepts. It is commonly understood that a single AI pipeline, with specified training and testing sets, can generate multiple models with varying degrees of generalizability. Underspecification refers to the pipeline’s inability to determine whether these models have integrated the structure of the underlying system by utilizing an independent test set that is distributed identically to the training set [5]. Radiomics provides a noninvasive way to evaluate and monitor tumor characteristics, such as temporal and spatial heterogeneity, by assessing the tumor and its microenvironment as a whole. For radiomics to become widely accepted in clinical practice as biomarkers, improvements, and standardization are necessary [2].

## 4. Types of Imaging 

Imaging has long been used to diagnose metastatic melanoma prior to the introduction of AI. Computerized tomography (CT) scans, single-photon emission computerized tomography (SPECT), and positron emission tomography (PET) imaging were used for staging and prognostication purposes. AI has the potential to drastically influence the way these imaging techniques are used.

### 4.1. AI on CT 

Several variables affect the accuracy of imaging features on CT, such as the specific acquisition and reconstruction parameters, as well as the quality of the contrast enhancement used during imaging [6]. Hence, these variables must be carefully controlled during data analysis to ensure accurate conclusions. Regions of interest found in CT, PET, and MRI are analyzed in a semi-automatic fashion. Experienced physicians must manually correct computer-aided outline detections found on open-source platforms such as 3DSlicer. These platforms typically offer online support and are regularly updated [2]. Therefore, consistent sampling is essential, as the variables have a significant impact on radiomics features [4]. A possible solution to this obstacle is using predefined fixed CT parameters for image acquisition in future prospective radiomics studies that can effectively reduce variability and improve the accuracy and reproducibility of radiomics results [7]. By using fixed CT parameters, such as consistent slice thickness, radiation dose, and reconstruction kernel, researchers can minimize the impact of technical factors on radiomic features and enable a more reliable comparison of imaging data across different patients and institutions [8]. This approach can also reduce the potential confounding effects of the scanner model, operator skill, and patient motion.

### 4.2. AI on MRI 

Most MRI equipment is non-standardized and varies from one manufacturer to another. MRI intensities vary due to this non-standardization, as well as different sequence types and acquisition parameters, resulting in inconsistency in image intensities between individuals and within individuals undergoing multiple MRIs over time [2]. This impacts radiomics features remarkably. MRI data preprocessing methods such as intensity normalization, bias field correction, and noise smoothing can help improve the quality of MRI and make the results of radiomics analysis more reliable and reproducible. Applying these preprocessing methods will make radiomics analyses more robust, and the results can be comparable across different studies and institutions [9].

### 4.3. AI on PET

The integration of PET images into radiation therapy workflows has proven to be highly beneficial. PET images allow for more precise tumor targeting, more consistent segmentation, and improved patient management and radiation therapy dose planning compared to CT or MRI [10]. Additionally, PET imaging can provide valuable information about the metabolic activity of tumors to help assess tumor aggressiveness and response to therapy [11]. The EANM Research FDG-PET/CT accreditation program has helped identify major calibration errors and reduce long-term inconsistencies of PET results, regardless of manufacturer or model [12]. 

### 4.4. AI in SPECT Imaging

Few reports have evaluated the use of SPECT imaging for radiation therapy due to its limited ability to provide quantitative information (photon attenuation, scatter, partial volume effect, and motion artifacts). However, these limitations are being addressed via the utilization of SPECT/CT acquisitions and quantitative image reconstruction [11] (Figure 1).

## 5. Detection and Diagnosis

Current standardized detection and diagnosis of cutaneous malignant melanoma involves dermoscopy with a biopsy of the lesion with appropriate dermatopathology and staging. Additionally, ultrasound has been used to help with detection and diagnosis. Recently, the detection and diagnosis of malignant cutaneous melanoma has involved the use of AI. Furthermore, the International Skin Imaging Collaboration has collected the largest set of images over the last five years to develop algorithms for AI-detected skin cancer [15]. New evidence has emerged suggesting that AI/ML can lead to better clinical decisions regarding diagnosis and detection, with the possibility of replacing human-based judgment. These studies have shown that AI/ML algorithms have performed better or as well as dermatologists [16].

One study recently compared the accuracy of human dermatologists to state-of-the-art AI technology in diagnosing melanoma-like pigmented skin lesions. The results showed that ML algorithms achieved a mean of 2.01 (95% CI 1.97 to 2.04; *p* < 0.0001) more correct diagnoses than the dermatologists [17]. Another recent study suggests that dermatologists’ confidence in the diagnosis and detection of cutaneous melanoma increased with the confirmation of AI-based deep neural networks, such as the convolutional neural network (CNN) [18].

There are other investigational imaging forms of detection and diagnosis of malignant cutaneous melanoma that involve the use of targeted molecular imaging using PET/CT with biomarkers underway. Fibroblast activation protein (FAP)-targeted PET imaging, as well as programmed death ligand-1 (PD-L1)-targeted PET imaging, have recently been shown to be extremely effective in the detection of various malignant cancers, including malignant melanoma. Furthermore, melanin imaging has shown utility in the early diagnosis of cutaneous melanoma, with many more immune-targeted therapies currently undergoing clinical trials [19]. PD-L1 targets the cell line in A375 melanoma cells transfected with the human PD-L1 gene, whereas the FAP targets are notorious for being ubiquitous in tumor cells, and several clinical trials are underway regarding the diagnostic results of using these two markers [19]. Additionally, the melanin targets are highly overexpressed in melanoma cells and have the potential to decrease the costs and shortcomings in the detection of micro melanomas [19]. There are currently specific limitations for the initial diagnosis of melanoma, such as tracer availability, histotype, and resolution limits that AI has the ability to improve via its ability to use targeted molecular imaging and enhanced algorithms.

## 6. Staging

Medical imaging is essential for tumor staging in patients with metastatic melanoma. Currently, TNM staging is standard practice for metastatic melanoma. The T stage is defined according to Breslow thickness and ulceration. The N stage is determined by regional node metastasis. The M category primarily focuses on the site of distant metastases, with the category definitions also including lactate dehydrogenase level [20].

CT scans, ultrasound, sentinel node biopsy, and SPECT-CT are the standard for evaluating nodal involvement. Other modalities, such as high- or ultra-high frequency ultrasound, are new to the field, with high resolution potentially enabling important staging information without the need for biopsy. An important predictor of survival in patients with melanoma is the number of tumor-involving lymph nodes [20]. ^18^F-FDG PET/CT has also been proven to have high performance in the diagnosis/detection of nodal metastases at initial staging [10,21].

Additionally, targeted molecular imaging represents a new strategy to identify diseases via PET that are not visualized by traditional imaging [19]. A new trend is forming wherein biomarkers equipped with AI features allow for the exploration of advanced tumor architecture, orientation, and histologic structures; these biomarkers can be used routinely for cancer staging once they become validated [22,23].

## 7. Prognostication

The current (eighth) edition of TNM (tumor, node, and metastasis) staging of cutaneous melanoma described by the American Joint Committee on Cancer (AJCC) is meant to be prognostic rather than predictive. The criteria are based on the characteristics of the primary tumor (thickness, ulceration, and mitotic rate), regional lymph node status, and the presence of distant metastases. The information from TNM staging is then combined to classify patients into AJCC prognostic stage (stage I-IV) groups. Studies have established a strong relationship between these features and patient survival [24]. Many other prognostic factors, including age, sex, sentinel lymph node tumor burden, mitotic rate, and circulating melanoma cells or tumor DNA (ctDNA), can strongly impact long-term patient outcomes but are not formal components of the AJCC system due to a lack of sufficient data. They are likely to influence future versions of the prognostic criteria. 

AI has been used on histological images of melanoma specimens to create computational algorithms that can predict disease-free survival. In one study, a CNN (convolutional neural network) was created using digital slides from 108 patients and tested on its ability to predict distant metastatic recurrences (DMRs) over 24 months on two validation sets of 104 and 51 patients. The AUCs (area under the curve) were 0.91 and 0.88 for the two sets, respectively, and the outputs also correlated with disease-specific survival [25]. Another study used support vector machine (SVM)-based statistical analysis on digital slides to predict sentinel lymph node (SLN) positivity. The model had a sensitivity and specificity of 77% and 94% using SVM and 93% and 69% using logistic regression, respectively [26]. Such studies are relatively sparse, and the use of AI in pathology is not yet well validated. More research is required to predict metastases, drug response, and survival time in melanoma. Nevertheless, AI holds tremendous potential in aiding the prognostication of cutaneous melanoma as an adjunct to TNM staging and reducing unnecessary sentinel node biopsies. Additionally, AI can incorporate important prognostic factors beyond the AJCC criteria and even identify novel factors such as genetic profiles, making it a promising tool for managing melanoma.

## 8. Treatment Planning and Treatment Delivery

While surgical excision can successfully treat most melanomas, patients with metastatic cutaneous melanoma often do not benefit from surgery due to the spread of cancer. Despite improvements in systemic therapies, the long-term outcomes for metastatic melanoma are still poor. Targeted therapies, such as BRAF and MEK inhibitors, have shown promising results in patients with specific gene alterations, but acquired resistance can limit their efficacy. Immunotherapy has demonstrated a durable response in approximately half of patients with metastatic melanoma, but severe autoimmune adverse events affect many patients [3]. Imaging has also played a crucial role in locating existing or metastatic diseases while also helping with delivery and treatment planning. The use of PET imaging agents is on the rise, with many biomarkers having great potential—including but not limited to FAP, melanin, MEK, nicotinamide, and benzamide [19]. Data suggest that melanin-targeted agents are more specific and advantageous compared to others, including FAP, MEK, and benzamide. This new technique can help address the various limitations seen in standardized imaging by increasing specificity and detecting lesions below the resolution of current imaging techniques by mapping tumor heterogeneity and providing more specific prognostic information [19]. Additionally, the majority of—if not all—cancer treatments, particularly those used to treat metastatic melanoma, have been validated thanks to a radiologic scanner. The best indicator of a drug’s efficacy remains overall survival; therefore, we need surrogate markers to determine progression-free survival, which is then extracted from the radiological evaluation of the scanners thanks to the criteria generally demonstrated by RECIST (response evaluation criteria in solid tumors) or iRECIST in the framework of melanomas being treated by immunotherapy [19].

AI can offer a potential solution to the increasing complexity of treatment delivery, representing a revolutionary advancement in the role of image-guided immunotherapy. Additionally, the use of AI imaging techniques can also help with more specific disease detection to allow for appropriate planning of immunotherapy and facilitate a more direct treatment delivery approach. They can also help identify adverse effects or disease progression in patients undergoing immunomodulatory therapy [23]. A potential example could be having a patient start one dose of immunotherapy and then repeat imaging with AI, informing whether the patient should continue the same drug or escalate to another. This could involve starting with nivolumab and relatlimab and then switching to a more toxic form, such as nivolumab and ipilimumab, depending on imaging results. Furthermore, AI tools will allow clinicians to adapt treatments earlier on and more reliably by deciphering the tumor immune microenvironment using medical images to recognize patterns and thus guide treatment delivery and planning [3].

## 9. Response Assessment

It is important to note that the current AJCC melanoma staging system likely underestimates survival outcomes, as it is based on data that predates the use of new highly effective therapies such as checkpoint inhibitor immunotherapy and molecularly targeted therapy. The criteria need to be readjusted with these contemporary treatments in mind. 

AI can be used to analyze cutaneous melanoma histology specimens to predict therapeutic responses to immune checkpoint inhibitors. Hu et al. demonstrated that a CNN, based solely on digital histological slides, could accurately predict response to immune checkpoint inhibitors in 54 melanoma cases, with an area under the curve (AUC) of 0.778. CNN correctly classified 65.2% of responders and 74.2% of non-responders [27]. In another study, a multivariate classifier effectively stratified patients into high and low risk for disease progression, with an AUC of 0.800. Patients classified as high risk had significantly worse progression-free survival than those classified as low risk (*p* = 0.02) [28].

Recent studies have demonstrated the effectiveness of radiomics and machine learning in predicting treatment response and overall survival in patients with advanced melanoma treated with immunotherapy. One study identified a radiomic signature discerned from conventional CT that reached an AUC of 0.92 for overall survival in a validation set of patients with melanoma treated with pembrolizumab [13]. Similar studies have demonstrated the effectiveness of CT-based radiomic signatures in predicting overall survival in patients with NSCLC treated with nivolumab [29]. Such signatures can be developed for other imaging modalities, such as FDG-PET scans. Seban et al. identified radiographic signature biomarkers involved in lymphoid tissue metabolism in the spleen and bone marrow to predict survival and response in patients with melanoma prior to anti-PD1 immunotherapy [23,30,31,32]. 

## 10. Treatment Toxicity Assessment

Traditional cancer treatments focus on targeting cell division [14]. Their effectiveness can be assessed based on the regression and shrinkage of tumors, which can best be measured by CT using Response Evaluation Criteria in Solid Tumors (RECIST) [33].

However, immunotherapy causes different patterns of tumor progression, resulting in a need for fundamental changes in the use of imaging modalities to study their beneficial and adverse effects [14]. First, pseudotumor progression is a pattern that results in a transient increase in tumor size [34]. Next is dissociated response, a mixture of decreases in the size of some lesions and increases and/or no change in size in others [35]. On the other hand, many individuals would advance from immunotherapy without any changes in the size of their lesions [14]. Abscopal response, which refers to tumor shrinkage in a site that is not directly targeted, is another form of response [36].

Additionally, immunotherapeutic treatments may adversely affect various organs, resulting in autoimmune toxicity such as pneumonitis and thyroiditis, known as immune-related adverse events (irAEs) [14,37]. Identifying these patterns can be challenging since irAEs can also cause cell swelling, making it hard to differentiate the impact of these treatments on tumor cells to determine whether treatment was effective or an adverse effect was skewing the information [14]. Furthermore, patients who respond well to treatments and have longer exposure duration may have a greater risk of experiencing irAEs [38]. Therefore, it is crucial to explore new imaging techniques to better understand the relationship between irAEs and treatment response [14].

Numerous studies have demonstrated the effectiveness of utilizing AI and radiomics, which transform images into quantitative data, to analyze various tumor responses to immunotherapy [3,14]. Although CT images are used more frequently due to their accessibility, AI can also use images derived from MRI and ^18^F-FDG PET [14,39]. AI and radiomics have the potential to assess survival rate, disease prognosis, treatment response, and irAEs [14,40,41]. Although there is reassuring information about AI, we still have a long way to go before it can be effectively applied in clinical settings [14].

Besides CT scans, some new imaging modalities may also be beneficial in assessing immunotherapy. New MRI techniques, such as perfusion-weighted, apparent diffusion, MR spectroscopy, and chemical exchange saturation transfer (CEST) MRI, have shown promise in the field of immunotherapy [14,42]. Moreover, it has been shown that ^18^F-FDG PET can detect irAE in patients with melanoma receiving immunotherapy even before the usual clinical presentations of irAE [43]. Another optimal imaging modality that may be used in the future is immune PET, which analyzes the immune context of tumors [14].

Furthermore, stereotactic radiosurgery (SRS), which delivers a high dose of radiation to lesions, is a treatment option for brain metastases [44]. Radiation necrosis (RN) is one of the most crucial adverse effects of these treatments and must be differentiated from tumor recurrence (TR) due to the need to treat these entities differently [44,45]. Compared to conventional imaging methods such as MRI with contrast, modalities such as perfusion MRI, MR spectroscopy, and PET have been shown to better distinguish between RN and TR; however, there is no sufficient sensitivity or specificity of these modalities in the clinical fields [46,47]. Many studies have shown the potential promising role of radiomics in differentiating between RN and TR [48,49]. The use of machine learning and radiomics has shown that using an imaging marker on conventional T1 MRI, which indicates the texture and spatial characteristics of the lesion, can differentiate between RN and TR [50].

## 11. Surveillance

Radiologic surveillance of melanoma has increased recently as a method to monitor patients for early recurrence and distant metastasis. Distant disease spread patterns for melanoma are typically hard to predict and are often atypical compared to other tumors. This has created inconsistencies in guidelines for the use of imaging surveillance, creating a space for AI-based therapies to establish a foothold. 

Basic radiology interventions such as CXR and ultrasound imaging have typically been used to detect the recurrence of metastatic disease. Ultrasound surveillance is usually superior to clinical examination alone, which often leads to false negatives; however, the long-term benefit of using ultrasound for surveillance is unknown. Thus, many patients who are negative for recurrence undergo unnecessary surgery or repeat scans. More modern techniques, such as CT and PET/CT, have become the preferred modality for surveillance. One study shows that PET/CT sensitivity was 65% and specificity was 99% for surveillance of distant metastasis, with similar results being reported in other studies [51]. Patients with metastatic melanoma typically undergo surveillance within 6–12 months following surgery using PET/CT. Of note, the use of PET/CT for the detection of brain metastasis is typically poor, which is of crucial importance due to the brain being a common site for metastasis. Instead, MRI is the preferred imaging modality for surveillance [51].

Increased surveillance using AI could result in earlier treatment for recurrence and improved early diagnosis of recurrence. Furthermore, AI could potentially help gauge the likelihood a radiographic abnormality is correctly identifying actual melanoma. This would then guide the decision of whether a biopsy is needed. There are currently several models that provide a prognostic benchmark for the use of biomarkers in metastatic melanoma. One example is using ML to predict the recurrence of bladder cancer post-cystectomy with greater specificity and sensitivity than prior imaging techniques [52]. Furthermore, recurrence prediction using ML with digital pathology has shown great promise with several cancers, including melanoma. Similarly, DL with PET scans is currently being used to predict local recurrence of cervical cancer, and applied ML to CT-derived radiomic features is starting to predict recurrence in various cancers. Of note, a recent paper published in 2022 outlines a new ML algorithm to predict recurrence risk using 36 clinicopathologic features. The most predictive features for recurrence were Breslow tumor thickness and mitotic rate. This study holds great promise in the field of AI-related surveillance of metastatic melanoma [53]. Imaging-based radiomics and DL modalities for cancer surveillance have rapidly grown in the last decade, further allowing for the application of metastatic melanoma and future recommendations for individualized surveillance and treatment [54]. 

Existing radiomics-based analyses focus on molecular and histologic classification using biomarkers from the primary tumor. There is great potential to use biomarkers with imaging such as PET, CT, and MRI using radiomics to enhance current clinical surveillance of metastatic melanoma. Further studies to explore the full potential of AI-based surveillance are needed.

## 12. Perspectives and Future Directions

Immune checkpoint inhibitors, such as anti-CTLA4, anti-PD1, and anti-LAG3, are currently the standard treatment options for cutaneous metastatic melanoma; however, further research on the best ways to combine and sequence these drugs is needed [55]. ^18^F-FDG PET/CT has shown accuracy not only in response monitoring but also in monitoring systemic immune response and detecting prognosis in an early stage [56]. There is a future need to provide a more accurate and systematic interpretation of ^18^F-FDG PET/CT by implementing new evolving immunotherapeutic strategies [23]. Furthermore, the use of ^18^F-FDG PET biomarkers with immunotherapy is in its early stages, and advancements in its clinical application are imperative in relation to better overall survival rates [30]. Other recent advances include the ability of radiomics applied to CT to non-invasively assess tumor heterogeneity that could be an indicator of response to immunotherapy [40]. Further standardization is needed to reflect changes in the tumor microenvironment to confirm this potential advancement [13]. 

Future directions include determining the total tumor burden by analyzing all measurable lesions and estimating overall survival using AI [57], which can extract tumor volume, heterogeneity, and shape [13]. Continued research in AI using imaging modalities such as CT, MRI, and PET—the current standards of care—could transform these tasks. The detection of molecular pathways that provide physiological information showing disease response and progression will become increasingly more specific. The increase in encouraging results using AI and radiomics supports further advancements in precision medicine with metastatic melanoma. Using biomarkers with AI can help predict outcomes and overall survival with greater accuracy, generalizability, and reproducibility. Additionally, the automatic extraction of data with AI is currently not being utilized due to time constraints; however, new research in the field will help mitigate this challenge. Continued clinical trials and ongoing innovation will help determine earlier on if treatments are working for a patient’s specific tumor characteristics and better predict overall survival [58].

## 13. Conclusions

The use of AI methods, such as radiomics and DL algorithms, is rapidly advancing the field of medical imaging and holds tremendous potential for improving the diagnosis and management of patients with metastatic melanoma. However, significant challenges remain, including underspecification and overfitting, which must be addressed to implement AI-based decision-making tools in clinical practice. Standardization and improvement in radiomics are necessary for it to become a widely accepted biomarker. Moreover, the choice of imaging modality significantly impacts the accuracy and reproducibility of radiomics analysis. Controlling variables during data analysis can make the results more robust and reliable, and the use of fixed parameters can minimize the impact of technical factors. In addition, AI has the potential to assist clinicians with prognostication of cutaneous melanoma by incorporating important prognostic factors beyond the AJCC criteria and reducing unnecessary sentinel node biopsies. AI in response assessment to contemporary treatments such as immune checkpoint inhibitors has also shown promising results. However, more research is required to predict metastases, drug response, and survival time in melanoma, and the AJCC criteria need to be readjusted with contemporary treatments in mind. In summary, advancements in AI provide hope for developing new therapeutic strategies to fight deadly diseases like metastatic melanoma, and careful consideration of imaging modality and variables during data analysis can make radiomics analysis more reliable and comparable across different studies and institutions.

## Figures and Tables

**Figure 1 diagnostics-13-03483-f001:**
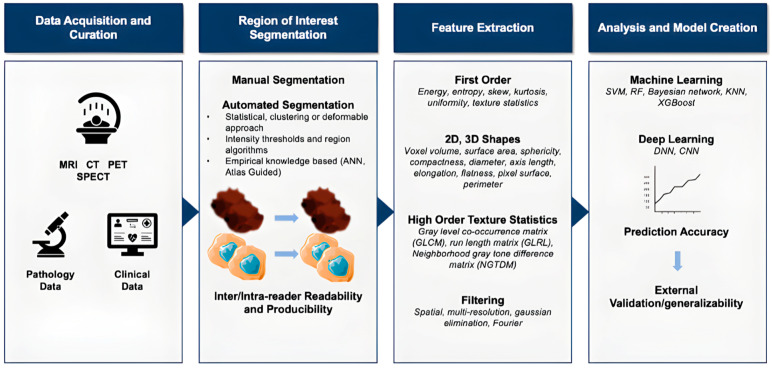
Overview of imaging modalities and methods that can be used with AI. MRI, SPECT, PET, and CT are all primarily involved with the diagnosis of metastatic melanoma. Integrating AI into imaging allows clinicians to find a segment of interest and then extract features to draw correlations and identify the most important features. AI models are created with these data and can be used to diagnose, prognosticate, treat, and monitor metastatic melanoma [2,4,11,13,14].

## Data Availability

All data are included in the manuscript.

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
