# Peer review of "Recent Advances in the Field of Artificial Intelligence for Precision Medicine in Patients with a Diagnosis of Metastatic Cutaneous Melanoma"

_diagnostics, 2023, doi:10.3390/diagnostics13223483_

Round 1

Reviewer 1 Report

Comments and Suggestions for Authors

This is an interesting review article reporting on the application of AI in patients with cutaneous melanoma. The article is well-written and encompasses a comprehensive evaluation of the literature. In section 5 (Detection and diagnosis) the authors should mention ultrasonography as a technique employed for lesions' evaluation along with dermoscopy. The authors are encouraged to check previous literature on the topic (see 10.1002/jum.16096).

Author Response

Thank you very much for your comments! I am eager to add them into our finished paper!

Reviewer 2 Report

Comments and Suggestions for Authors

The manuscript submitted by Higgins et al. offers a comprehensive examination of the pivotal role of medical imaging in cancer management, with a particular emphasis on the escalating issue of malignant cutaneous melanoma. It effectively underscores the significance of immunotherapy and targeted protein inhibitors as leading treatment strategies, surpassing previously limited options. Moreover, the manuscript presents a compelling exploration of the burgeoning field of artificial intelligence (AI) research in cutaneous melanoma, demonstrating its potential to transform various aspects of clinical practice, from diagnosis to treatment planning. In light of its comprehensive coverage and well-organized content, the manuscript is deemed acceptable for publication with only minor comments.

1. To enhance the clarity and readability of the manuscript, it is recommended to define acronyms the first time they are mentioned by providing the full term followed by the acronym in parentheses. This practice will ensure that readers can easily grasp the meaning of the acronyms and follow the text more effectively.

Author Response

Thank you very much for your feedback! I look forward to adding your comments into our paper.